# Cellular coordination underpins rapid reversals in gliding filamentous cyanobacteria and its loss results in plectonemes

Jerko Rosko[1†], Rebecca N Poon[1†], Kelsey Cremin[1†], Emanuele Locatelli[2†], Mary Coates[1], Sarah JN Duxbury[1], Kieran Randall[1], Katie Croft[3‡], Chantal Valeriani[4], Marco Polin[3,5], Orkun S Soyer[1*]

[1]School of Life Sciences, University of Warwick, Coventry, United Kingdom; [2]Department of Physics and Astronomy, University of Padova, Padova, Italy; [3]School of Physics, University of Warwick, Coventry, United Kingdom; [4]Departamento de Estructura de la Materia, Física Termica y Electronica, Facultad de Ciencias Físicas, Universidad Complutense de Madrid, Madrid, Spain; [5]Instituto Mediterráneo de Estudios Avanzados, IMEDEA, Esporles, Spain

**\*For correspondence:**
O.Soyer@warwick.ac.uk

[†]These authors contributed equally to this work

**Present address:** [‡]Medical Research Council, Weatherall Institute of Molecular Medicine, University of Oxford, Oxford, United Kingdom

**Competing interest:** The authors declare that no competing interests exist.

## eLife Assessment

Using microscopy experiments and theoretical modelling, the authors present **convincing** evidence of cellular coordination in the gliding filamentous cyanobacterium Fluctiforma draycotensis. The results are **fundamental** for the understanding of cyanobacterial motility and the underlying molecular and mechanical pathways of cellular coordination.

**Abstract** Cyanobacteria are key contributors to biogeochemical cycles through photosynthesis and carbon fixation. In filamentous, multicellular cyanobacteria, these functions can be influenced through gliding motility, which enables filaments to localise in response to light and also form aggregates. Here, we use the aggregate-forming species *Fluctiforma draycotensis* to study gliding motility dynamics in detail. We find that filaments move in curved and straight trajectories interspersed with reorientation or reversal of direction. Most reversals take a few seconds, but some take substantially longer, resulting in a long-tailed distribution of stoppage times. Mean filament speeds range around a micron per second with a relatively uniform distribution against filament length, implying that all or a fixed proportion of cells in a filament contribute to movement. We implement a biophysical model that can recapitulate these findings. Model simulations show that for filaments to reverse quickly, cells in a filament must achieve high coordination of the direction of the forces that they generate. To seek experimental support for this prediction, we track individual cells in a filament. This reveals that cells' translational movement is fully coupled with their rotation along the long axis of the filament, and that cellular movement remains coordinated throughout a reversal. For some filaments, especially longer ones, however, we also find that cellular coordination can be lost, and filaments can form buckles that can twist around themselves, resulting in plectonemes. The experimental findings and the biophysical model presented here will inform future studies of individual and collective filament movement.

## Introduction

Cyanobacteria are key contributors to global primary production and oxygen generation (*Hartmann et al., 2014*; *Hamilton et al., 2016*). They display high diversity and are adapted to a wide range of habitats from the soil crust to freshwater and the ocean (*Hamilton et al., 2016*; *Cohen and Rosenberg, 1989*). Within this diversity, some cyanobacterial species display multicellularity, existing as filaments composed of multiple individual cells. Many of these filamentous cyanobacteria display a specific type of motility, known as gliding motility (*Cohen and Rosenberg, 1989*; *Risser, 2023*; *Schuergers et al., 2017*; *Wilde and Mullineaux, 2015*). Gliding motility enables filaments to position themselves in response to light (*Kurjahn et al., 2024*; *Lamparter et al., 2022*; *Campbell et al., 2015*; *Nultsch and Häder, 1988*) and also to form aggregates (*Shepard and Sumner, 2010*; *Sato et al., 2014*; *Duxbury et al., 2022*), thereby enhancing photosynthesis and additional functions such as nitrogen fixation and iron acquisition (*Kessler et al., 2020*).

Gliding motility is a specific form of bacterial motility that occurs on surfaces without significant deformation or use of flagella (*Hoiczyk, 2000*; *McBride, 2001*; *Mauriello et al., 2010*; *Nan, 2017*; *Henrichsen, 1972*). In single-celled bacteria, studies in the species *Flavobacterium johnsoniae* and *Myxococcus xanthus* have shown that gliding motility involves membrane-bound protein complexes moving in helical patterns across and around the cell body (*Sun et al., 2011*; *Nan et al., 2011*; *Nakane et al., 2013*; *Nan et al., 2013*). These membrane proteins are suggested to reach so-called focal adhesion points, at cell-surface contact points, where they convert their motion into axial forces that translate the cell and rotate it around its long axis (*Nan, 2017*; *Faure et al., 2016*). Mutational studies in these species, as well as filamentous cyanobacteria, have also identified proteins involved in the biosynthesis and secretion of a sugar polymer, so-called slime, to be essential for gliding motility (*Hoiczyk, 2000*; *McBride, 2001*; *Mauriello et al., 2010*; *Nan, 2017*; *Risser and Meeks, 2013*; *Zuniga et al., 2020*). In filamentous cyanobacteria, helically organised surface fibril proteins (*Halfen and Castenholz, 1970*; *Hoiczyk and Baumeister, 1995*; *Adams et al., 1999*; *Read et al., 2007*) and junctional protein complexes (*Hoiczyk and Baumeister, 1998*) were identified and suggested to link with motility and slime secretion (*Halfen and Castenholz, 1970*; *Hoiczyk and Baumeister, 1998*). Type IV pili machinery is also implicated in filamenteous cyanobacteria gliding, both in *Nostoc punctiforme*, where only differentiated filaments called hormogonia display gliding motility (*Risser, 2023*), and in other species (*Lamparter et al., 2022*; *Khayatan et al., 2015*; *Zuckerman et al., 2022*; *Cho et al., 2017*).

Despite these ongoing efforts on identifying the molecular mechanisms of force generation, the dynamics of gliding motility in filamentous cyanobacteria remains poorly characterised. In particular, it is not clear how filamentous cyanobacteria achieve coordination of cellular propulsive forces during gliding (*Wilde and Mullineaux, 2015*). A better understanding of motility dynamics can provide insights into cellular coordination and inform molecular studies of propulsion. Study of the motility dynamics at the filament level is also relevant to population-level observations, such as aggregate formation. Current theoretical explanations for these observations make use of specific assumptions about filament behaviour (*Pfreundt et al., 2023*; *Faluweki et al., 2023*), which can be better justified, or ruled out, upon better understanding of movement dynamics.

Here, we study gliding motility dynamics in a filamentous cyanobacterium *Fluctiforma draycotensis*, capable of gliding motility and extensive aggregate formation (*Duxbury et al., 2022*). We focus specifically on characterisation of movement dynamics, rather than elucidating their underlying molecular mechanisms. To this end, we used extensive time-lapse microscopy under phase, fluorescent, and total internal reflection (TIRF) modalities. We found that gliding motility involves movement on curved or straight trajectories interspersed with rapid reorientation or reversal events and with mean speeds independent of filament length. We were able to recapitulate these findings with a physical model of filament movement, which predicts a high importance for cellular coordination to achieve rapid reversals. We were able to confirm this prediction by tracking individual cells during a reversal. The single-cell analysis also revealed a direct coupling of translational and rotational movement, which can generate torsional forces during reversals. In support of such forces, we found that longer filaments tend to readily buckle and twist upon themselves to form plectonemes. The formation of plectonemes is associated with loss of cell-to-cell coordination during reversals, leading to filament ends moving independently. The presented findings quantify individual filament movement and provide a physical model of gliding motility that is consistent with experimental observations.

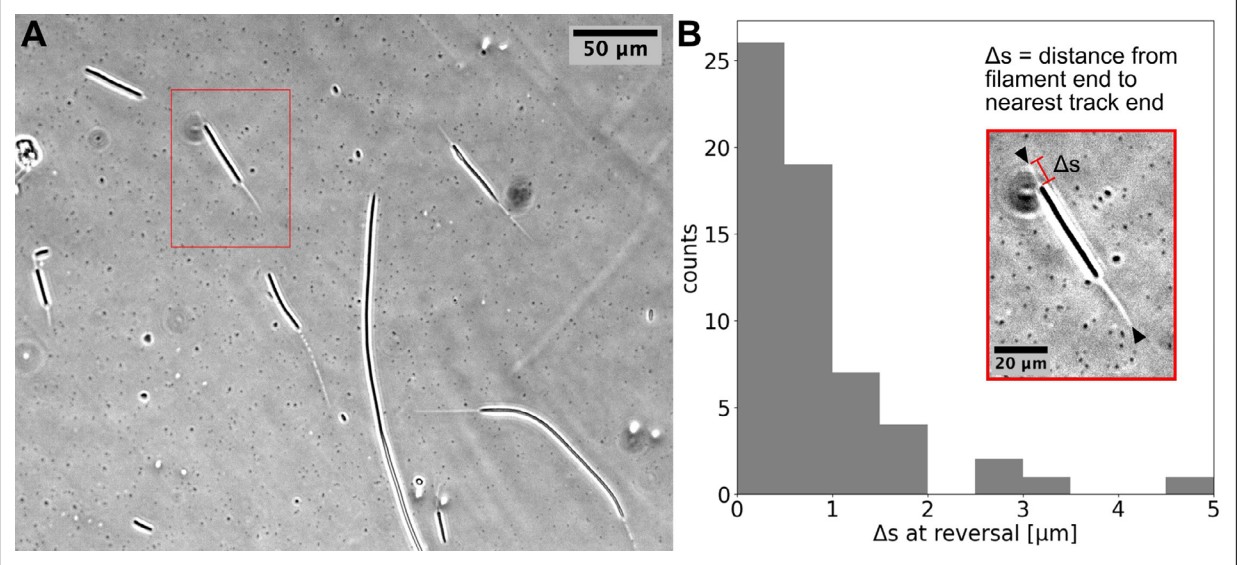

**Figure 1.** Cyanobacterial filaments move on `tracks' on agar. (**A**) Phase contrast image of filaments on an agar pad, showing `tracks' associated with moving filaments. (**B**) Distribution of distance between the end of the filament and the closest track end ($\Delta s$) at reversals (see Inset). Data is from multiple filaments ($n = 14$), each showing several reversals, resulting in 61 data points. Approximately 70% ~of the $\Delta s$ values are less than the measurement error ($\approx 1$). Inset: the single filament and track highlighted in (**A**) with a red box. Track ends are indicated by black arrowheads.

The online version of this article includes the following video and figure supplement(s) for figure 1:

**Figure supplement 1.** Reversal frequency as a function of different variables.

**Figure 1—video 1.** Filament moving under 1.5% agarose sandwich.

https://elifesciences.org/articles/100768/figures#fig1video1

**Figure 1—video 2.** Single filament in track on 2.5% agarose.

https://elifesciences.org/articles/100768/figures#fig1video2

**Figure 1—video 3.** Single filament on a circular trajectory on 1.5% agarose.

https://elifesciences.org/articles/100768/figures#fig1video3

**Figure 1—video 4.** Several filaments in track on 2.5% agarose.

https://elifesciences.org/articles/100768/figures#fig1video4

Together, they will inform future studies on the molecular and physical mechanisms of force generation and collective behaviour of filaments.

## Results

### Filament movement is interspersed with stoppage events, whose duration has a long-tailed distribution

Using time-lapse microscopy, we observed filament movement on glass slides and under agar pads (see *Materials and methods*). We found that filaments move on straight or curved trajectories that are interspersed with reorientation or reversal of direction. Under agar, it was possible to distinguish a region of different phase contrast, which aligned with the trajectories of the filaments, forming a 'track' (*Figure 1A* and *Figure 1—videos 1; 2*). We note that it is possible that the tracks are associated with the secreted slime, see *Discussion* section. Lone filaments' trajectories were mostly confined to these tracks, involving a stoppage and reversal at each end of the track (*Figure 1B*), while filaments entering a circular track did not readily reverse (*Figure 1—video 3*). We also observed filaments sometimes reversing without reaching track ends, but when this happened, it was usually at a consistent location (*Figure 1—video 4*). Thus, the observed tracks under phase microscopy constitute a confining space that bounds filament movement and dictates reversal frequency (*Figure 1—figure supplement 1*). On glass, filament movement still displayed a go-stop-go pattern, but stoppages led to both reversal and reorientation of direction, and as a result, trajectories were less well-defined. Reversals and

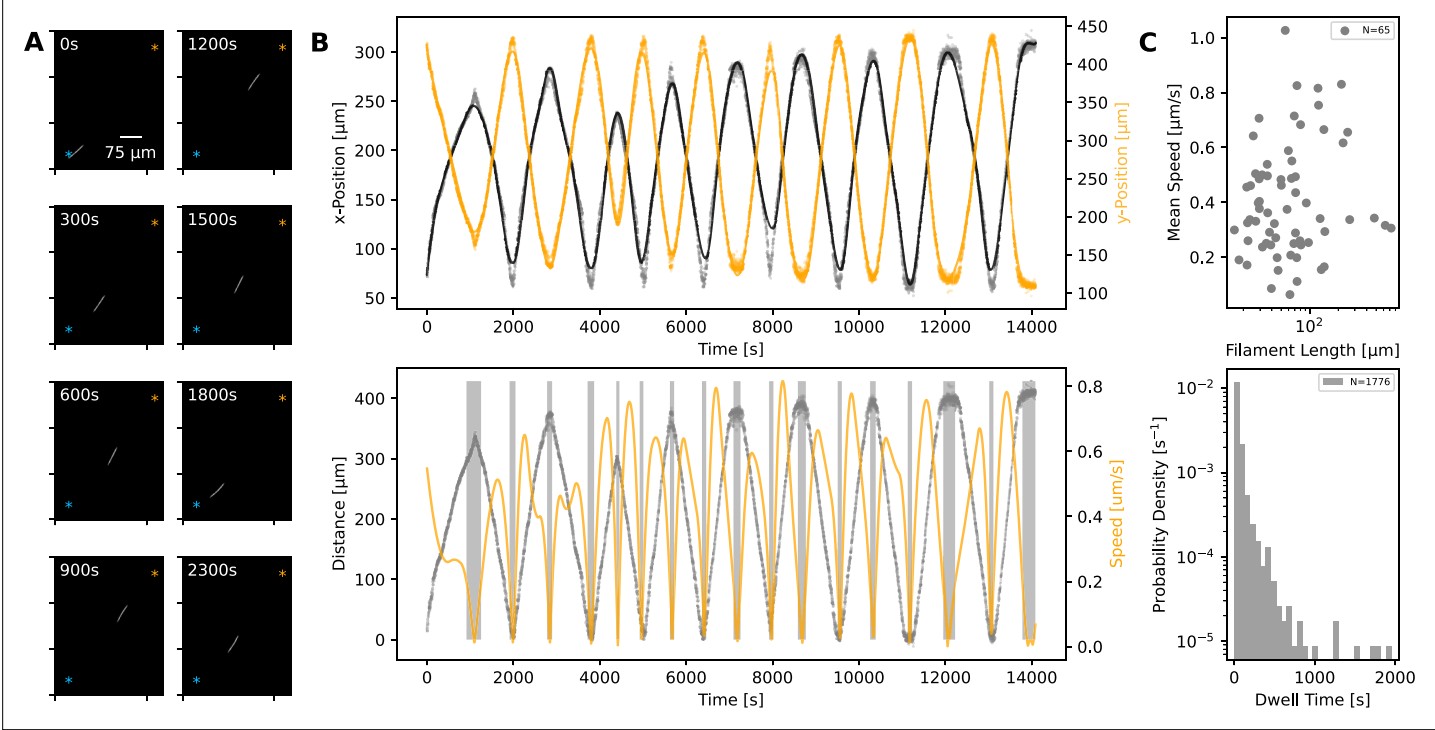

**Figure 2.** Statistics of filament motion on agar. (**A**) A single filament shown at different time points of its movement under an agar pad. Time-lapse images were captured at 2 s intervals using fluorescence microscopy. The scale bar shown on the first image applies to all subsequent ones. The extreme points of the trajectory across the time lapse are marked with blue and orange asterisks on each image. (**B**) Top: X- and Y-coordinates of the filament's centre throughout the recorded time-lapse. The points show observations, while the line shows a spline fit to this data. Bottom: The distance (grey) between filament centre and one of the extreme ends of its trajectory, shown with blue asterisk on panel A, and filament speed (orange) throughout the time-lapse. The speed is calculated from the spline fitted to the x- and y-coordinates shown in the top panel. Grey backdrop regions indicate time points with speed below a set threshold, indicating reversal events. (**C**) Top: Mean filament speed from 65 different filaments observed under agar, plotted against filament length. Bottom: Distribution of dwell times, as calculated from independent reversal events. For the same analyses for observations on glass, see (*Figure 2—figure supplement 2*).

The online version of this article includes the following video and figure supplement(s) for figure 2:

**Figure supplement 1.** Example filament trajectories on glass.

**Figure supplement 2.** Summary statistics for filament motion on glass.

**Figure supplement 3.** Distribution of dwell times of leading (head) and trailing (tail) ends of filaments.

**Figure supplement 4.** Summary statistics for stopping frequency.

**Figure 2—video 1.** Filaments moving across glass slide.

https://elifesciences.org/articles/100768/figures#fig2video1

reorientations were associated with a deceleration-acceleration of filaments (*Figure 2A, B*, *Figure 2—figure supplement 1*). This decrease in speed can either result from external counter forces or loss of propulsive force, for example through filaments reaching the end of a track under agar or detaching from the surface on glass (see (*Figure 1—video 1* and *Figure 2—video 1*)). Mean filament speeds ranged around 0.5 µms⁻¹ and displayed a relatively uniform distribution against filament length on agar (*Figure 2C*) and a weak positive correlation with filament length on glass (*Figure 2—figure supplement 2*). Since filament speed results from a balance between propulsive forces and drag, these observations of no or positive correlation between filament speed and length show that all (or a fixed proportion of) cells in a filament contribute to propulsive force generation.

We can consider the reversal/reorientation events of a filament as akin to tumbling events seen in flagella-based bacterial motility. To this end, we were interested in the distribution of time spent during reversals, which we call the 'dwell time'. Analysing over 1700 reversal events across 65 filaments, we found that dwell times display a long-tailed distribution (*Figure 2C*). On glass, we found a similar distribution from 1434 reversal/reorientation events (*Figure 2—figure supplement 2*). Most

reversals happened within a few seconds, while a minority involved significant time spent stationary. We found that there was no clear difference between the dwell time of leading and trailing ends of the filament (*Figure 2—figure supplement 3*). Taken together, these findings show that filament reversals are mostly rapid events that do not involve delays across the filament. The filaments on agar reverse with a frequency that is inversely proportional to the filament length (which is in turn proportional to the track length; see *Figure 1—figure supplement 1*). In contrast, we find that the frequencies of reversals on glass do not show a correlation with the filament length and are narrowly distributed (see *Figure 2—figure supplement 4*). These findings are in line with the idea that tracks on agar are defined by filament length and dictate reversal frequency, resulting in strong correlations between reversal frequency, track length, and filament length. On glass, filament movement is not constrained by tracks, and we have a specific reversal frequency independent of filament length.

## A biophysical model captures filament movement dynamics and highlights a key role for cell-to-cell coordination for achieving fast reversals

As summarised in the introduction, the molecular basis of propulsive forces is not fully understood. However, the presented quantification of motility dynamics highlights three key observations that can be used to develop a plausible biophysical model of filament movement. Firstly, and most obviously, cells in a filament remain physically linked during motion and therefore must exert mechanical forces onto each other. Secondly, all or most cells contribute to propulsive force generation, as seen from a uniform distribution of mean speed across different filament lengths (*Figure 2C*, *Figure 2—figure supplement 2*). Thirdly, and finally, reversals are associated with reduction in speed (or stoppage) and usually happen within seconds.

We implement the first two observations by modelling cells in a filament as beads connected by springs, with each cell capable of generating propulsion and exerting pull and push forces onto others (see *Figure 3A* and *Materials and methods*). The third observation shows that cells have a mechanism to alter the direction of their propulsive force. To this end, we model reversals as resulting from a stochastic cellular signalling pathway that is linked to mechano-sensing of external and internal forces acting on a given cell (see *Materials and methods* and *Equations 3; 6*). This modelling choice effectively assumes that cells have an intrinsic reversal rate, but adjust this in such a way to reduce any compression or extension that they perceive. In addition, and to account for cellular sensing dynamics, we introduce a memory (i.e. a refractory period) that limits cells' ability to switch force direction if they have recently done so (see *Equations 3; 4* in *Materials and methods*).

Given this model, we simulated the movement of filaments in a one-dimensional space and in the presence of an external force field. The latter allows us to implement a reduction of speed and analyse filament movement on a defined track, as observed experimentally for the movement under agar (*Figure 2*). For appropriate choice of parameters, we found that this model can produce sustained back-and-forth movement (*Figure 3B* and *Figure 3—video 1*). We also run simulations in the absence of an external force field, more closely mimicking the case on glass. Here, again, the model was able to reproduce reversals and experimentally observed dwell time distributions (*Figure 3—figure supplements 1 and 2*).

The ability of the model to present reversals can be understood from the way it implements mechano-sensing at the cell level. On simulations mimicking the agar, when leading cells reach the track ends, the external force field causes an increase in their tendency to reverse the direction of their propulsive force (*Figure 3B*). When this happens, trailing cells are compressed by both those in front and behind them, and, if cell-cell coupling is strong enough, are forced to reverse as well. Thus, the mechano-sensing creates a signal that travels across the filament and sustains a coordinated reversal event. The same dynamics can also be created from stochastic reversal of a few individual cells, and as such, we also observed reversals in the model simulations without the filament reaching the track ends, or without an external force field for some parameter values (*Figure 3B*, *Figure 3—figure supplements 1 and 2*).

To better understand how the key model parameters affect reversals, we run simulations across a wide range of parameter values (*Figure 3C and D*). To quantify reversal dynamics, we considered two measures that are based on cell-to-cell coordination and extent of reversals. The reversal-related measure, $M$, gives the ratio of the number of actual reversals over the estimated number of reversals

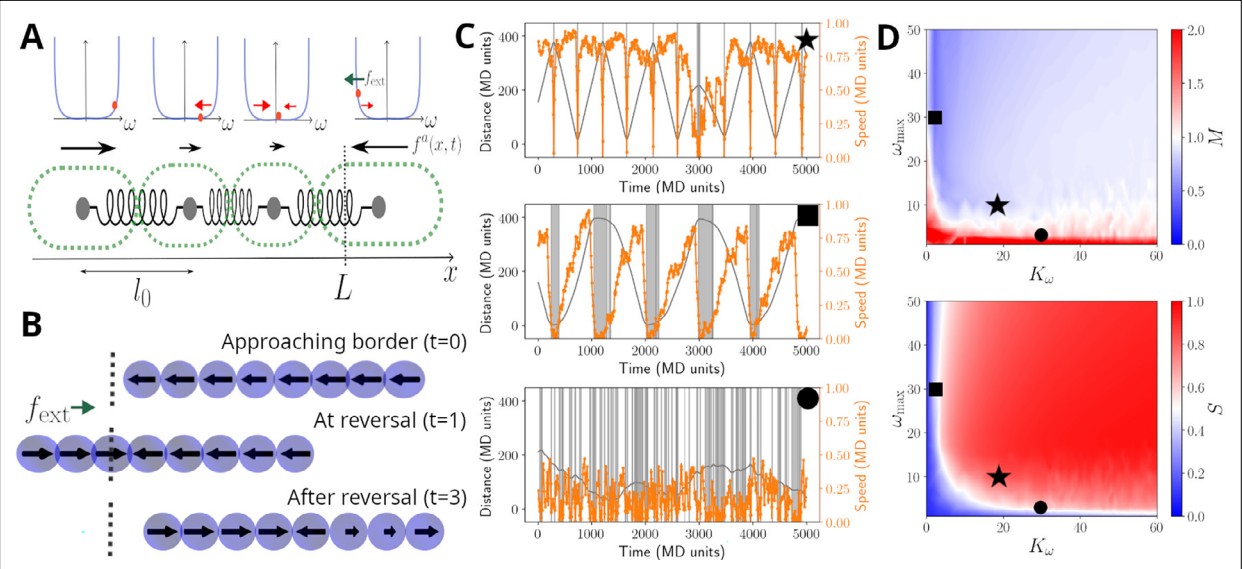

**Figure 3.** Model sketch, examples, and performance. (**A**) A cartoon of the biophysical model (see *Materials and methods*). Cells are modelled as beads, connected by springs, with preferred rest length $l_0$. Cells self-propel with a propulsion force $f_a$. It is assumed that cells regulate the direction of $f_a$, and this regulation is modelled by a function $\omega$, which includes a self-regulatory element, implemented as a confining potential $U_\omega$ (shown as blue lines in the panels above cells). The value $\omega_i$ of each cell is represented with an orange dot, and it's affected by random fluctuations, a mechanical feedback from neighbouring cells ($V_\omega$, red arrows), and an external signal $f_{\text{ext}}$, present at the ends of the track $L$ (green arrow; see *Equation 5*). (**B**) Example of reversal process: the snapshots show a coordinated filament approaching the border (top); after reaching it, the closest cells reverse their propulsion under the action of $f_{\text{ext}}$ (centre). This prompts the rest of the cells to reverse and the filament coordinates again to travel in the opposite direction (bottom). (**C**) Simulated trajectories (grey lines) and absolute speed value (orange lines) as a function of time, presented as in *Figure 2B*, for $N_f = 70$ units in a track of length $L/l_0 = 400$. The different panels display a typical trajectory for a well-behaved filament (top, $K_\omega = 20$, $\omega_{\text{max}} = 10$), a filament with little cell-to-cell coupling (centre, $K_\omega = 2$, $\omega_{\text{max}} = 30$) or with little memory (bottom, $K_\omega = 30$, $\omega_{\text{max}} = 1$). The grey bands highlight reversal events and their duration. (**D**) Contour plots of the synchronisation $S$ (top) and of the reversal efficiency $M$ (bottom) as a function of the cell-to-cell coupling $K_\omega$ and of the cell memory $\omega_{\text{max}}$. See Materials and methods section for the definition of $S$ and $M$. Black symbols highlight the systems showcased in panels (**C**).

The online version of this article includes the following video and figure supplement(s) for figure 3:

**Figure supplement 1.** Distribution of stop/dwell times for simulated filaments on 'glass', that is without external forces.

**Figure supplement 2.** Distribution of stop/dwell times for $N_f$=100, $K_\omega = 25$ and several values of $\omega_{\text{max}}$, for simulations without external force field.

**Figure supplement 3.** Simulated distribution of dwell times.

**Figure supplement 4.** Number of reversals $n_r$ against the number of expected reversals $n_r^{(e)}$ as observed in experiments under agar (**A**) or in simulations (**B**).

**Figure supplement 5.** Average reversal frequency for stop-reverse (blue) and stop-go (orange) events as a function of $K_\omega$ for $N_f = 100$ and different values of memory parameter, $\omega_{\text{max}}$, for simulations without external force field.

**Figure supplement 6.** The relation of reversal frequency (**A**) and stop frequency (**B**) with filament length and memory parameter (i.e., in the $N_f - \omega_{\text{max}}$ plane) at fixed $K_\omega = 15$, for simulations without external force field.

**Figure supplement 7.** The reversal frequency (**A**) and median filament velocity (**B**) against Nf, and $\omega$ max set using $\omega$ max $\propto$ N$-0.23$ f (at fixed K$\omega$ =15) for simulations without external field.

**Figure 3—video 1.** Simulation movie of an active filament with a parameter set, resulting in coordinated reversals.
https://elifesciences.org/articles/100768/figures#fig3video1

**Figure 3—video 2.** Simulation movie of an active filament with a parameter set, resulting in erratic reversals.
https://elifesciences.org/articles/100768/figures#fig3video2

obtained by considering an idealised filament, that has a defined mean speed and reverses deterministically and instantaneously at track ends (*Equation 7*). Thus, a value of $M \approx 1$ can indicate regular and rapid reversals at track ends, although it can also indicate simply a high reversal frequency. The coordination measure, $S$, gives the extent of homogeneity of cellular propulsive forces across a filament (*Equation 9*), with a maximum value of 1, corresponding to completely synchronised cell motion. Evaluating $S$ and $M$ together allows us to better quantify coordination in reversal dynamics along a track. We find that both measures are sensitive to the strength of the cell-to-cell coupling and the memory

parameter (*Figure 3C*). Indeed, when both parameters are reduced, reversals become more erratic, with filaments undergoing many reversals and individual cells having opposing propulsive force directions (*Figure 3B* and *Figure 3—video 2*). As expected from these results, these two parameters also affect the shape of the dwell time distribution resulting from model simulations (*Figure 3—figure supplement 3*). For sufficiently high cell-to-cell coupling and memory parameters, simulated filaments reverse rapidly, and primarily at track ends, and cellular propulsive force directions remain homogeneous throughout their movement (*Figure 3B*). To explore if real filaments' movement conforms to such expected dynamics arising from high cell-to-cell coupling, we quantified the number of expected reversals using individual filament and track lengths, and the observed mean filament speed from all observations. We found that for the high majority of the filaments, the number of observed reversals matches closely with the expected number of reversals, in qualitative agreement with the model simulations performed with sufficiently high cell-to-cell coupling parameter (*Figure 3—figure supplement 4*).

## Filament movement shows high cell-to-cell coordination and coupling of translation with rotation

Taken together, the experimental and model results presented so far show that most reversals are rapid events, and to achieve them, the filaments should have high cell-to-cell coupling in their movement. To explore this model-highlighted idea of cellular coordination and to better understand what happens during a reversal at the single cell level, we undertook Total Internal Reflection Fluorescence (TIRF) microscopy, which illuminates only an ≈ 200-nm-thick band of the sample and allows observation of surface features (*Figure 4A–B*, top panels). This allowed us to clearly identify individual cells and their septa with neighbouring cells, which we tracked to quantify individual cell movement. We found that cell movement across the filament mostly stays coordinated during a reversal, in terms of speed and direction of movement (*Figure 4A*, bottom panels, and *Figure 4—video 1*).

In addition to identifying individual cells, TIRF microscopy revealed membrane-bound protein complexes localised predominantly near the poles of each cell (*Figure 4A*). We have also tracked these complexes and found that their trajectories match 2-dimensional projections of a three-dimensional helical trajectory, showing that filaments rotate as they translate (*Figure 4D*, and *Figure 4—video 1*). The angle $\theta$, that the trajectories make with the long axis of the filament shows a normal distribution with a mean of 21.2°, while the angular speed of the protein complexes is well correlated with the linear velocity of the filament (*Figure 4E and F*). These findings indicate either helical force generation or a screw-like structure on the cell exterior that causes a strong coupling between rotation and translation. It is interesting to note in this context that staining filaments with fluorescently labelled Concanavalin A (see *Materials and methods*), which binds to slime, results in a helical pattern along the filament long axis (*Figure 4—figure supplement 1*). In time-lapse TIRF movies, cellular protein complexes can be seen moving through this fluorescence pattern, suggesting that the slime surrounds the filaments as a tubular structure, fixed in position relative to filament movement (*Figure 4—video 2*). The tubular nature of slime is further supported by scanning electron microscopy (SEM) images, which show features around the filaments that look like collapsed slime tubes (*Figure 4—figure supplement 2*).

## Reversals in longer filaments can lead to buckling and plectoneme formation, causing loss of cellular coordination

The above findings show that filaments display a high level of cell-to-cell coordination, supporting the model result that cellular coordination is one of the requirements for rapid reversals. In addition, we find a strong coupling between rotation and translation. Thus, it is possible that any differences in cell movement, for example, the leading cells initiating reversal earlier, could lead to torsional forces developing along the filament. Supporting this, we note that cellular speeds display a steep acceleration immediately after a reversal (*Figure 4B and C*), a characteristic observed in all filaments imaged using TIRF. In a few filaments, we have also observed indications of leading cells initiating reversal earlier than trailing ones (*Figure 4—figure supplement 3* and *Figure 4—video 3*). To further explore this possibility of de-coordinated movement dynamics along the filament and the emergence of associated torsional forces, we sought to analyse longer filaments with the expectation that such dynamics are expected to arise more readily in longer filaments. We found that longer filaments,

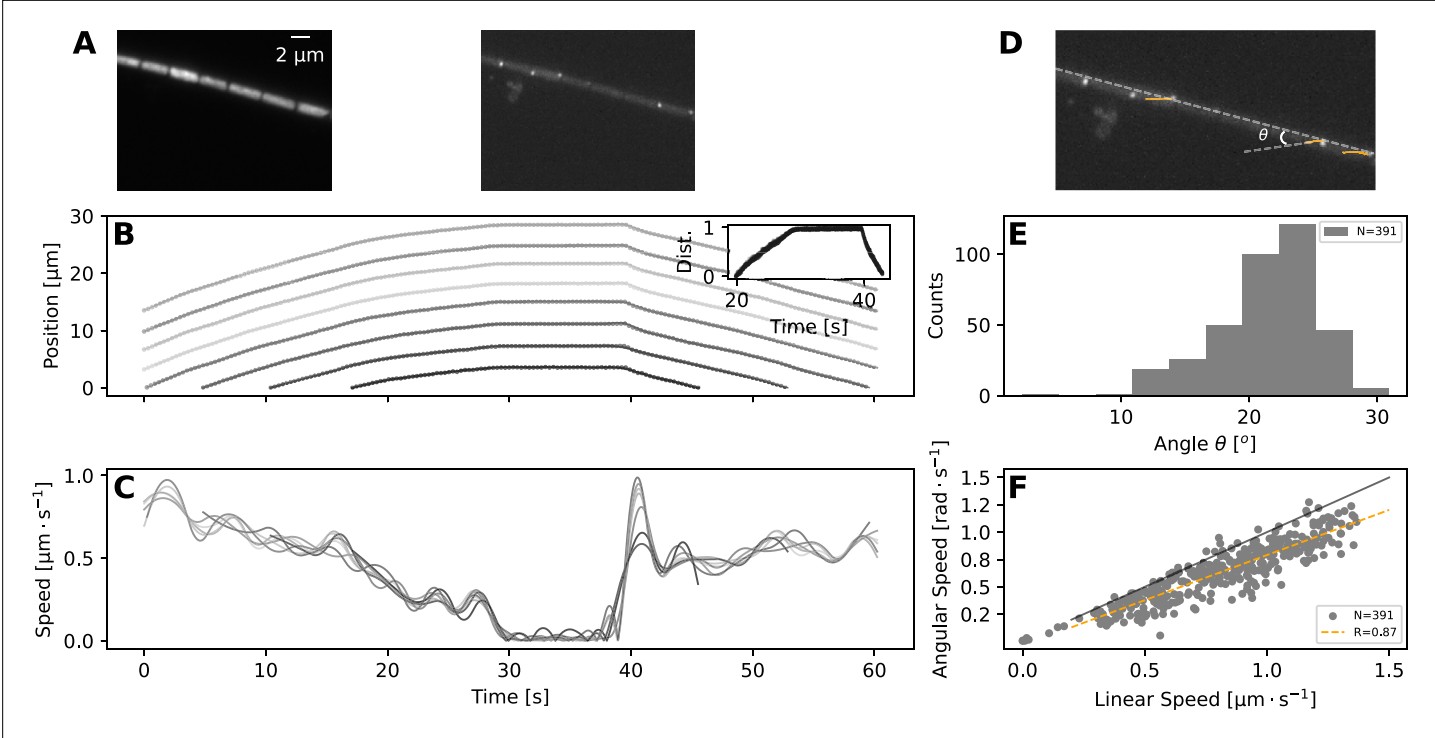

**Figure 4.** TIRF imaging reveals coupled rotation and translation. (**A**) Individual images from a TIRF microscopy time-lapse, obtained with excitation using 473 nm laser. The left and right panel show images with emission filters centered at $625nm$ and $525nm$, respectively. Note that these images show a thin ($\approx 200$nm) section of the cell membrane. (**B,C**) show respectively the position and speed of individual cells during this TIRF time-lapse movie (*Figure 4—video 1*). Different cells' trajectories and speed are shown in different shades of grey. The inset on panel (**B**) shows the normalised distance travelled by each cell, revealing high coordination in their movement. (**D**) The same TIRF image as shown on the right of (**A**), annotated with the trajectories of some of the membrane-bound protein complexes (orange lines) and the angle of this trajectory with the long axis of the filament; $\theta$. (**E,F**) show respectively the distribution of the angle $\theta$; and the relation between rotational speed of the protein complexes and the linear speed of the filament. The black and orange lines in (**F**) show the expected diagonal and the correlation fit (see legend) between linear and angular speed. Both angle and speed measurements are collected from 11 filaments and 391 trajectories.

The online version of this article includes the following video and figure supplement(s) for figure 4:

**Figure supplement 1.** TIRF microscopy images of filament bundles.

**Figure supplement 2.** Scanning electron microscopy (SEM) images of the *F. draycotensis* cyanobacteria filaments.

**Figure supplement 3.** De-coordinated dynamics following reversal.

**Figure supplement 4.** Distribution of filament lengths that exhibit buckling or plectonemes, shown for filaments observed on a glass slide (n=50), or sandwiched under agar (n=11).

**Figure 4—video 1.** TIRF imaging of a filament undergoing a reversal.
https://elifesciences.org/articles/100768/figures#fig4video1

**Figure 4—video 2.** TIRF imaging of stacked filaments in a stained slime sheath.
https://elifesciences.org/articles/100768/figures#fig4video2

**Figure 4—video 3.** TIRF imaging of a filament undergoing a reversal, where fluorescence shows the protein complexes on the surface.
https://elifesciences.org/articles/100768/figures#fig4video3

where observed, readily display buckling, and sometimes these buckled loop regions twist upon themselves and form plectonemes. Almost all of the 61 observed cases of buckling and plectoneme formation involved filaments longer than 400 µm (*Figure 4—figure supplement 4*).

Given that plectonemes can form, we next wanted to understand cell-to-cell coordination during such events. While it was not possible to capture a filament with TIRF microscopy, during a plectoneme formation event, we were able to record whole-filament movies at low magnification. A representative case is shown in *Figure 5* (and *Figure 5—video 1*), where we were able to analyse movement dynamics for each end of the filament. We found that the filament initially moves in a coordinated manner, on a well-defined trajectory, and the two ends of the filament being coordinated in speed (*Figure 5B*).

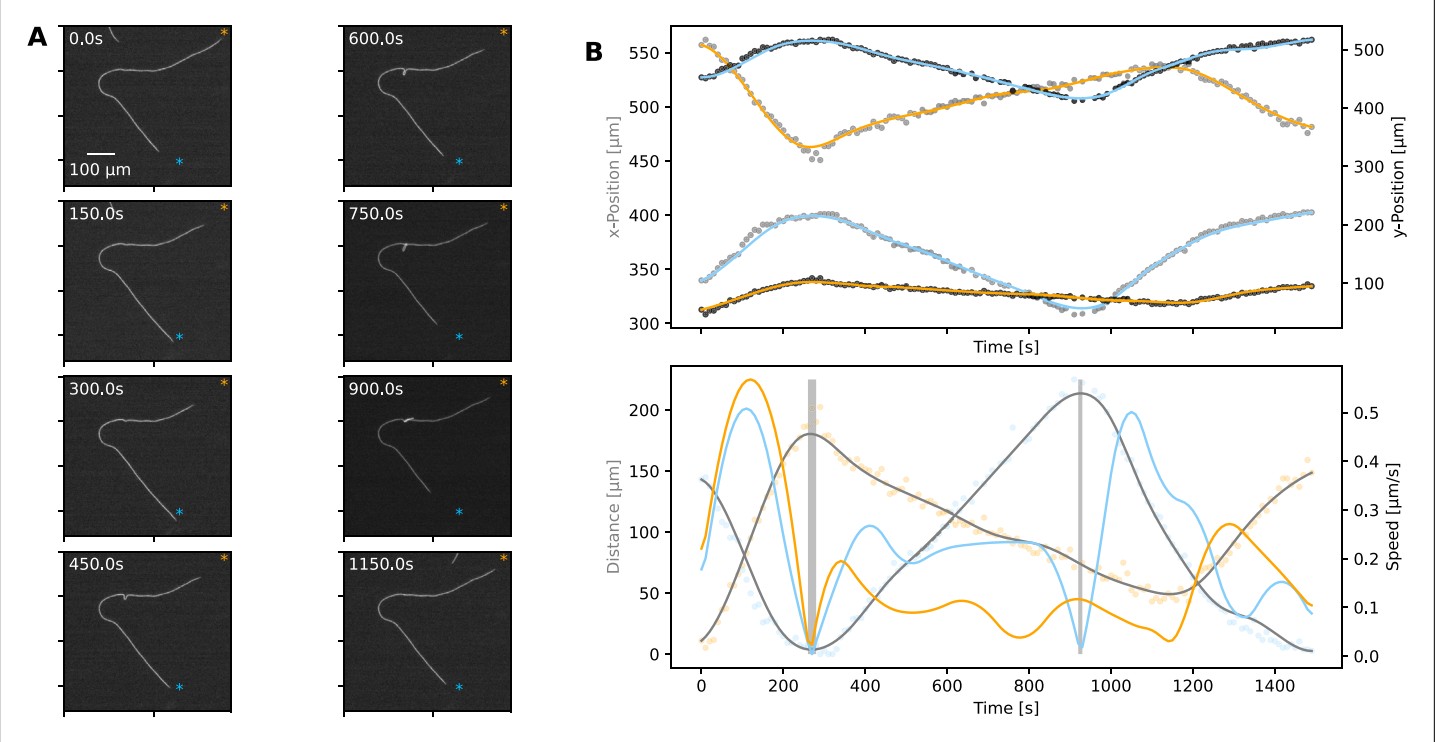

**Figure 5.** Plectoneme formation. (**A**) A filament forming a plectoneme during movement under agar, shown at different time points as indicated on each panel. (See *Figure 5—video 1* for full video.) The scale bar shown on the first image applies to all subsequent ones. The orange and blue asterisks indicate the end-points of the filament's trajectory. (**B**) Top: The x- and y-coordinates of the two ends of the filament over time. Data points for the x- and y-coordinates are shown in grey and black, respectively. Lines are spline fits to the data, and their orange and blue colours indicate the filament end that stays close to the corresponding trajectory end-point shown on the images in panel A. Bottom: Speed and distance of each end, respective to the end-point of the trajectory close to them. Speed is shown as blue and orange lines, calculated from the respective position data shown in the top panel. Distance is shown in orange and blue data points, indicating the filament end that stays close to the corresponding trajectory end-point shown on the images in panel A, while grey lines show spline fits. The shaded areas indicate the reversals. Note that the second reversal involves only the end that stays close to the blue asterisk on panel A.

The online version of this article includes the following video(s) for figure 5:

**Figure 5—video 1.** Buckling filament under 1.5% agarose.

https://elifesciences.org/articles/100768/figures#fig5video1

**Figure 5—video 2.** Filament twisted in the middle, forming plectonemes on glass.

https://elifesciences.org/articles/100768/figures#fig5video2

**Figure 5—video 3.** Filaments forming plectonemes on glass.

https://elifesciences.org/articles/100768/figures#fig5video3

With the start of buckling and plectoneme formation, however, this coordination is disrupted, with one end of the filament moving faster than the other and finally reversing independently of the other end (*Figure 5B*). This de-coordination seems to resolve towards the end of the recorded movement sequence, when the plectoneme resolves (*Figure 5* and *Figure 5—video 1*). This representative case exemplifies the fact that buckling, and more specifically plectoneme formation, correlates with de-coordination of cells within a filament and independent movement of filament ends. We have observed more extreme cases of such dynamics on glass, where plectonemes correspond with de-coordination in filament movement (*Figure 5—video 2*). We also observed that plectonemes can get "dragged" and sometimes entangle other filaments (*Figure 5—video 3*), a dynamic process that could be relevant for initiating aggregation of multiple filaments.

## Discussion

Here, we analysed the motion dynamics of the filamentous, gliding cyanobacterium *F. draycotensis*. We show that gliding involves movement of filaments on curved or straight trajectories, interspersed with reorientation or reversal events. Filament speed decreases and then increases around a reversal event, with mean speed around 1 μm/s, independent of filament length. We find that the time spent in reversal displays a long-tailed distribution, with most reversals taking few seconds, but some involving considerable time of filaments being stationary. We show that these experimental observations can be recapitulated by a physical model that assumes individual force generation by each cell, mechanical coupling among cells, and mechano-sensory (coordinated) control of force direction. This model suggests that such coordination of propulsion direction among cells in a filament is important for being able to achieve fast reversals. In line with this insight, we find that cells in a filament remain highly coordinated during reversals and that their translational movement is coupled with rotation along the long axis of the filament. Following from the latter observation, we find that longer filaments can readily buckle, and buckled regions can twist upon themselves to form plectonemes. The formation of plectonemes results in de-coordination of cellular movement and independent dynamics for filament ends. We hypothesize such dynamics to be linked with the loss of sensory coupling among cells and de-coordination of the direction of their propulsive forces. Taken together, these results provide a useful characterisation of gliding motility, which will inform future studies on both molecular mechanisms of force generation and collective behaviours of filaments.

The presented model focussed on capturing filament reversals and cell coordination during these, for which a 1D representation and mechanical coupling was sufficient. The mechanical coupling, and in particular sensing of external force fields implemented in this model, implies the presence of mechano-sensing. To this end, several cyanobacterial genomes are shown to harbour homologues of the *E. coli* mechano-sensory ion channels MscS and MscL (*Johnson et al., 2021*), and we have identified similar MscS protein sequences in *F. draycotensis*. Furthermore, *F. draycotensis*, as with other filamentous cyanobacteria, has genes associated with the type IV pili, which are implicated in the surface-based motility of filamentous cyanobacteria (*Risser, 2023*) and in mechano-sensing in other species (*Gordon and Wang, 2019*; *Kühn et al., 2021*). All of these factors suggest the possibility of mechano-sensing in these cyanobacteria.

In terms of future model development, we would highlight the need for expanding into 2D or 3D models to better represent rotational and elastic bending forces and plectoneme formation. Lacking these features, the current model was still able to capture reversal events and dwell times. More detailed models, however, might be needed to fully capture reversal frequency, which we found to be independent of filament length on glass, where filament motion is not constrained. The presented 1D model requires a dependence of cellular memory on filament length to reproduce this observation (*Figure 3—figure supplements 5–7*), which could be possible depending on molecular mechanisms of motility.

In terms of molecular mechanisms of motility and propulsive force generation, our findings that rotational and translational movement are closely coupled in *F. draycotensis* and that slime presents itself as a helical tubular structure around filaments are highly relevant. Rotation, as well as its absence, and surface fibrillar arrays have been commented on in several species of gliding, filamentous cyanobacteria, but the rotation-translation coupling was not quantified before (*Halfen and Castenholz, 1970*; *Read et al., 2007*; *Zuckerman et al., 2022*). Cellular rotation has also been highlighted as a key dynamical feature in the propulsive force generation in gliding, single-celled bacteria (*Nan, 2017*; *Faure et al., 2016*). Taken together, these observations point to the possibility that rotation-translation coupling and propulsive force generation in filamentous cyanobacteria involves helical tracks surrounding the cell and either motor complexes moving on them, as suggested for single-celled, gliding bacteria (*Nan, 2017*; *Faure et al., 2016*) or pili being pushed across them. Recent studies from *Nostoc* species indicate propulsion to result from pili extension (*Wilde and Mullineaux, 2015*; *Khayatan et al., 2015*), but whether that species presents filament rotation or fibrillar arrays is unclear (*Read et al., 2007*).

Photoresponse in the motility behaviour is shown in several gliding filamentous cyanobacteria (*Kurjahn et al., 2024*; *Lamparter et al., 2022*; *Campbell et al., 2015*; *Nultsch and Häder, 1988*), but a fully quantitative analysis of these at the single filament level is currently lacking. In this context, an interesting observation we made in this study is the presence of fluorescent, membrane-bound

protein complexes. We hypothesize that these complexes are associated with light sensing, since our tracking of them in TIRF microscopy always coincided with the reversal of filaments. Thus, it is possible that the wavelength at which they are excited (437 nm laser used in TIRF microscopy) is also acting as a sensory signal for reversals. This possibility will be explored in a future study, along with further characterisation of any phototaxis behaviour in *F. draycotensis*.

Gliding motility in filamentous cyanobacteria is commonly associated with aggregate or macro-scale structure formation. These formations are relevant in the context of physiological functions of the cyanobacteria and their associated microbial communities, as shown in the case of *F. draycotensis Duxbury et al., 2022*. Similarly, in the case of the ocean-dwelling species of the *Trichodesmium* genus, aggregate formation is linked with the biogeochemically relevant processes of iron acquisition (*Kessler et al., 2020*). The results presented here show that de-coordination of cellular forces within a filament can lead to the formation of plectonemes, which can entangle multiple filaments together. We also note that loss of motility results in the loss of aggregates in *F. draycotensis* cultures (*Duxbury et al., 2022*). Thus, further studies of motility and plectoneme formation can elucidate how aggregate and macro-scale formations develop and inform their physical modelling.

Given that gliding motility is observed in phylogenetically diverse bacteria, it is possible for it to be underpinned by different molecular mechanisms and present different motion dynamics. The inter-connectivity of propulsion and slime secretion is difficult to disentangle purely by molecular approaches, and the development of physical models solely based on identified proteins can be limited. In particular, drastically different physical models can be constructed from observed molecular constituents. We advocate coupling molecular studies with detailed analysis of motion dynamics, as quantification of these can provide an additional framework to constrain and test different models, ultimately providing a more complete understanding of gliding motility and associated collective behaviours.

## Materials and methods
### Sampling and Culturing
Cultures of *F. draycotensis* were grown in medicinal flasks, in volumes of 30 mL BG11 +with added vitamin mix as described previously (*Duxbury et al., 2022*). Cultures were kept under visible light irradiance of $\approx 30$ µM·photons·m$^{-2}$s$^{-1}$, across a repeating 12 hr light/12 hr dark cycle. Samples were obtained from cultures aged between 4 and 8 weeks, by withdrawing 0.5 mL of material containing several pieces of biofilm, transferring the aliquot into a 1.5 mL microfuge tube and vortexing for 1 min to break up the material. If the mixture was not homogeneous, additional mixing was performed using a 1 mL mechanical pipette.

### Sample preparation for imaging
Prior to imaging, collected inoculate was diluted into fresh BG11 + media with vitamin mix to a desired dilution (typically 1:50). For microscopy performed with $1.5 - 2\%$ agarose pads (*Rosko et al., 2022*), 3 µL of the homogenised diluted sample was added to the pad. The droplet was then left to dry for 10–15 min. Where stated, the agarose pad was flipped onto a 22×50 mm cover glass, sandwiching the sample between the agarose and the glass. Otherwise, the sample was left on top of the agarose, and the pad was placed in a cover glass bottomed dish (MatTek/CellVis) for microscopy.

As described, some samples were imaged inside a millifluidic chamber device (Ibidi µ-Slide III 3D Perfusion) or on glass slides, using a 1.5 ×1 cm Gene frame (Thermo Scientific) as a spacer. For glass slide samples, the Gene frame sticker was placed on the standard glass slide. 200 µL of diluted sample was added to the frame, and the chamber was sealed with a cover glass.

Regardless of sample preparation method, all prepared samples were left to settle for 30–60 min, being kept on a laboratory bench under room temperature and illumination.

### Imaging conditions
Light microscopy was performed on an Olympus IX83 microscope, using a CoolLED pE-300 illumination system. Imaging was performed using a CoolSnap-HQ2, with an imaging interval of 1–2 s or longer. The stage is enclosed in an environmental chamber and stabilised at 25° C. Phase imaging was performed with a low exposure rate ($\approx$5ms). Chlorophyll fluorescence was measured through a

Chroma DSRed filter cube, with a recorded excitation light power of 17.4 mW and an exposure time of 100ms. TIRF imaging was performed on a standard NanoImager system (ONI) using a 473 nm excitation laser and emission filters centered at 525 nm and 625 nm.

## Image analysis

Images were analysed with custom-written software in Python. Analyses involved tracking individual filaments, cells, or membrane complexes on each cell, across time-lapse images. In the case of filament tracking, movies of a larger field of view were cropped to focus on single filaments, and these were processed by fitting a contour on the filament and deriving a mid-point spline from this. By repeating this process across each image in a time-lapse, it was possible to track the position of middle, trailing, and leading ends of the filament and their speed. In the case of longer time-lapse movies, any drift in the images was stabilised using stationary filaments within the larger field of view. The $\Delta s$ values shown in *Figure 1* were measured from relevant frames of phase-contrast time-lapse movies, using ImageJ, for multiple reversals of multiple filaments. All image analysis code used is made available on the Github repository (copy archived at *Soyer and Locatelli, 2025*).

## Biophysical model

The model code is available at the Github repository. We model cyanobacteria as out-of-equilibrium filaments. For simplicity, we consider only a one-dimensional model, but extension to higher dimensions is possible. The filament is composed by $N_f$ beads that represent cells in a cyanobacteria filament. The approach to model the filament as connected cells with springs between them is similar to several other physical models of active filaments and polymers (*Isele-Holder et al., 2015*; *Winkler and Gompper, 2020*; *Bianco et al., 2018*). The position of bead is denoted by $x_i$ ($i = 1 \ldots N_f$). Beads are connected with their nearest neighbours, along the filament, by harmonic springs. Each bead follows an over-damped equation of motion, given by;

$$\dot{x}_i = \frac{1}{\gamma} \left( -\frac{dV}{dx_i} + f_i^a \cdot s_i(x,t) \right) + \xi_i(t) \tag{1}$$

where $\gamma$ is the friction coefficient and $\xi_i(t)$ is a Gaussian white noise, following $\langle \xi_i(t)\xi_j(t') \rangle = 2k_B T/\gamma \delta_{ij}\delta(t - t')$, $T$ being the temperature. The potential energy $V(x)$ reads

$$\beta V(x) = \sum_{i=1}^{N_f-1} \frac{1}{2}\beta K((x_{i+1} - x_i) - l_0)^2 \tag{2}$$

where $\beta = 1/k_B T$ is the inverse of the thermal energy ($k_B$ is the Boltzmann constant) and $K$ is the spring constant, representing the elastic response of the cell upon compression or extension. The rest length $l_0 = 1$ represents the unperturbed (average) length of a cell.

The remaining term in *Equation 1* represents the beads' self-propulsion, that is, there is a force $f_i^a(x,t) = f_i^a \cdot s_i(x,t)$, generated by the bead itself, that causes it to move in the positive or negative direction. The $f_i^a$ is defined as a constant in time, while $s_i(x,t)$ sets the 'active state' of bead at position $x$ and time $t$. To model cellular direction change, we introduce a function, $\omega_i(x,t)$, describing the internal 'state' of each cell and dictating its active state by the relation $s_i(x,t) = \tanh(\omega_i(x,t))$. Notice that $s_i(x,t)$ is a continuous variable, such that $-1 < s_i(x,t) < 1$. Thus, a cell can be propelled in the positive ($s_i(x,t) \approx +1$) or negative ($s_i(x,t) \approx -1$) $x$ direction, or it can be inert ($s_i(x,t) \approx 0$). From a biological perspective, $\omega_i(x,t)$ abstracts the behaviour of a cellular signalling network that integrates external and internal inputs for the cell and sets its propulsive direction. This idea is motivated by the fact that all studied unicellular gliding bacteria incorporate directional change, and individual cells in filamentous cyanobacteria are reported to change the location of their Type IV pili apparatus across the cell poles upon reversals (*Wilde and Mullineaux, 2015*; *Khayatan et al., 2015*).

We define $\omega_i(x,t)$ as integrating several inputs, as follows:

$$\dot{\omega}_i(x,t) = \frac{1}{\gamma_\omega} \left( -\frac{dU_c}{d\omega_i} - \frac{dV_\omega}{dx_i} + f_{ext}(x,t) \right) + \eta_i(t) \tag{3}$$

where $\gamma_\omega$ is a friction coefficient, $\eta_i(t)$ is a Gaussian white noise, $U_c$ is a confinement potential, $f_{ext}(x,t)$ is an (adimensional) external force field, and $V_\omega$ is a mechano-sensing (adimensional) potential. The

confinement potential is a mathematical abstraction that allows us to maintain the output of $\omega_i(x,t)$ in a finite interval. This potential is given by:

$$U_c(\omega) = \begin{cases} \frac{1}{2}(|\omega| - \omega_{\max})^4 & |\omega| > \omega_{\max} \\ 0 & \text{otherwise,} \end{cases} \tag{4}$$

where the parameter $\omega_{\max}$ determines the extreme values that $\omega_i(x,t)$ can attain. As such, this parameter acts as a memory term, controlling a refractory period where cells would not switch direction; when given terms in *Equation 3* push a cell in one direction, $\omega_{\max}$ will determine the time when their effects will start changing $\omega_i(x,t)$. The noise term captures any fluctuations, either internal to the cellular signalling networks or caused by external noise (thermal or otherwise). It ensures that the individual cells can also turn on or off or switch direction spontaneously. The noise term $\eta_i(t)$ follows $\langle \eta_i(t)\eta_j(t') \rangle = 2D_\omega \delta_{ij}\delta(t - t')$, with $D_\omega = k_B T_\omega / \gamma_\omega$ a diffusion coefficient, and $T_\omega$ being the temperature that characterises the magnitude of the fluctuations of $\omega$ in absence of external stimuli.

Following the experimental observations, we postulate the existence of an external, mechanical stimulus that keeps filaments on a well-defined trajectory on agar, that is, a track. For simplicity, we assume that the track is fixed and extends from $x/l_0 = 0$ to $x/l_0 = L$. The external stimulus field is given by the following functional form:

$$f_{\text{ext}}(x,t) = \begin{cases} f_{\text{ext}} & \text{if } x < 0 \\ -f_{\text{ext}} & \text{if } x > L \\ 0 & \text{else} \end{cases} \tag{5}$$

where $f_{\text{ext}}$ is an adimensional constant. Given *Equation 3*, the action of the external field drives the auxiliary variable to positive/negative values. When considering filaments on glass, we simply set this external stimulus to zero.

The potential $V_\omega$ encodes the mechano-sensitivity of each cell to their neighbours and is given as;

$$V_\omega(x) = \sum_{i=1}^{N_f - 1} \frac{1}{2} K_\omega((x_{i+1} - x_i) - l_0)^2 \tag{6}$$

where the quantity $K_\omega = f_{\text{ext}}/l_0 + \delta K_\omega$ is an effective spring constant. The function $V_\omega$ encodes the feedback to a mechanical stress (compression / extension) applied to each cell, compelling the direction of self-propulsion so that this stress is relieved.

Notice that, within the model, $f_{\text{ext}}$ essentially encodes the basal strength of the response of the cells to stimuli, either extra- or inter-cellular. The mechano-sensing can be more ($\delta K_\omega > 0$) or less ($\delta K_\omega < 0$) relevant than the external stimulus; naturally, we impose $\delta K_\omega > -|f_{\text{ext}}|$, that is, there cannot be a positive feedback to compression.

## Simulation details

We integrate the equations of motion *Equation 1* using a stochastic predictor-corrector algorithm, while the equations *Equation 3* are integrated using the Euler-Maruyama algorithm. In both cases, we employ an integration time step $10^{-3}\tau \leq \Delta t \leq 5 \cdot 10^{-3}\tau$, $\tau$ being the unit of time. As mentioned above, we set the average length of an unperturbed cell as the unit of length and set the unit of time as the diffusion time of a single inactive cell $\tau = l_0^2/2D_0$. For convenience, we set the unit of energy as the active 'work' $\epsilon = f^a l_0 = 1$; as such, we fix $f^a = 1$ throughout the simulations. We further fix the friction coefficient to $\gamma = 1\epsilon\tau/l_0^2$. We assume that the motion of the filament is dominated by its self-propulsion, i.e. $f^a l_0 \gg k_b T$ *and* we set $k_B T = 0.01\epsilon$.

Concerning the parameters that enter in the evolution of $\omega$, the scale of the 'energy' should be set here by $k_B T_\omega$. However, we set $k_B T_\omega = 0.1$ for convenience; we find that reasonably smooth reversals can be obtained for such a choice, maintaining the other parameters within the same order of magnitude. As such, we fix $f_{\text{ext}} = 10$.

## Movement characterisation

The analysis of experimental and simulated trajectories focuses on the dwell time and the number of reversals. The dwell time is computed as follows: the trajectory is smoothed via a spline, and the velocity is computed via numerical differentiation. A dwell time (or dwell event) is defined as the interval within which the velocity remains, in absolute value, below 15% of its maximal value.

A reversal corresponds to a change of sign of the centre of mass velocity of the filament after dwelling; we name $n_r$ the number of reversals counted along a trajectory. Similar to reorientations in experiments on glass, we also identify stop-continue events when removing the boundaries: here, the centre of mass velocity does not change sign after dwelling. We define two quantities: the reversal efficiency $M$ and the reversal rate $\nu$. The reversal efficiency aims to capture the relation between the 'ideal' and observed number of reversals, hence a value of one indicates expected behaviour of a filament on a defined track. Mathematically, it is defined as;

$$M = \frac{n_r}{n_r^{(e)}} = \frac{n_r}{\tau_{\mathrm{m}}/\tau_r^{(e)}} \tag{7}$$

with $\tau_r^{(e)} \approx (L - N_f l_0)/v$ and $\tau_{\mathrm{m}}$ the measurement time (either in experiments or simulations). The quantity $n_r^{(e)}$ is the number of reversals one would expect from an ideal filament, that reverses smoothly and deterministically in a negligible time and then travels at constant speed. In simulations, $v = f^a/\gamma = 1 l_0/\tau$ while in experiments, we estimate $v \approx 1$. The quantity $\tau_r^{(e)}$ is, thus, the ideal time between two reversals, that is, the time the centre of mass would take to travel from its initial condition (after a reversal) to the track boundary. By definition, this quantity pertains only to filaments confined between the track's ends. Conversely, the reversal rate can be defined also without the track boundaries: it is simply defined as the number of reversals over the measurement time

$$\nu = n_r/\tau_{\mathrm{m}} \tag{8}$$

Finally, we define a syncronisation index $S$ as

$$S = \left\langle \left| \frac{1}{N_f} \sum_i s_i \right| \right\rangle \tag{9}$$

where the average is performed over time or over the different realisations. Essentially, for each conformation, we compute the (instantaneous) average propulsion direction of each cell; as we are interested in the overall synchronisation of the filament regardless of its direction, we take the absolute value.

## Acknowledgements

We thank Douglas Risser and Jonasz Slomka for insightful comments on an earlier version of this manuscript. We acknowledge the help of imaging facility manager Ian Hands-Portman with microscopy setups and Nicole Robb for allowing access to her groups' TIRF microscopy. We would also like to acknowledge the University of Warwick Electron Microscopy RTP for assistance in the research described in this paper. MP acknowledges the fact that IMEDEA is an accredited 'Marìa de Maeztu Excellence Unit' (grant CEX2021-001198, funded by MCIN/AEI/10.13039/501100011033).

# Additional information

## Funding

| Funder | Grant reference number | Author |
| --- | --- | --- |
| Gordon and Betty Moore Foundation | 10.37807/GBMF9200 | Jerko Rosko<br>Rebecca N Poon<br>Kelsey Cremin<br>Mary Coates<br>Sarah JN Duxbury<br>Kieran Randall<br>Katie Croft<br>Orkun Soyer |
| Ministero dell'Istruzione, dell'Università e della Ricerca | Rita Levi Motalcini | Emanuele Locatelli |
| Ministerio de Asuntos Económicos y Transformación Digital, Gobierno de España | IHRC22/00002 | Chantal Valeriani |
| Ministerio de Asuntos Económicos y Transformación Digital, Gobierno de España | PID2022-140407NB-C21 | Chantal Valeriani |
| Leverhulme Trust | RPG-2018-345 | Marco Polin |
| Ministerio de Ciencia e Innovación | CEX202-001198 | Marco Polin |

The funders had no role in study design, data collection and interpretation, or the decision to submit the work for publication.

## Author contributions

Jerko Rosko, Rebecca N Poon, Emanuele Locatelli, Formal analysis, Investigation, Methodology, Writing – review and editing; Kelsey Cremin, Formal analysis, Methodology; Mary Coates, Sarah JN Duxbury, Kieran Randall, Chantal Valeriani, Methodology; Katie Croft, Investigation, Methodology; Marco Polin, Conceptualization, Supervision, Writing – review and editing; Orkun S Soyer, Conceptualization, Formal analysis, Supervision, Funding acquisition, Writing – original draft, Writing – review and editing

## Author ORCIDs

Rebecca N Poon ![ORCID] https://orcid.org/0000-0001-6764-5258
Kelsey Cremin ![ORCID] https://orcid.org/0009-0000-4028-7155
Emanuele Locatelli ![ORCID] https://orcid.org/0000-0002-5507-1282
Katie Croft ![ORCID] https://orcid.org/0000-0002-6083-6569
Marco Polin ![ORCID] https://orcid.org/0000-0002-0623-3046
Orkun S Soyer ![ORCID] https://orcid.org/0000-0002-9504-3796

Reviewer #1 (Public review): https://doi.org/10.7554/eLife.100768.3.sa1
Reviewer #2 (Public review): https://doi.org/10.7554/eLife.100768.3.sa2
Reviewer #3 (Public review): https://doi.org/10.7554/eLife.100768.3.sa3
Author response https://doi.org/10.7554/eLife.100768.3.sa4

# Additional files

## Supplementary files
MDAR checklist

### Data availability

Data, models and analysis code are made available in a dedicated Github repository under MIT license (copy archived at *Soyer and Locatelli, 2025*).

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
