## [Editor Report · eLife Assessment]

Using microscopy experiments and theoretical modelling, the authors present **convincing** evidence of cellular coordination in the gliding filamentous cyanobacterium Fluctiforma draycotensis. The results are **fundamental** for the understanding of cyanobacterial motility and the underlying molecular and mechanical pathways of cellular coordination.

---

## [Referee Report · Reviewer #1 (Public review)]

Summary:

The authors use microscopy experiments to track the gliding motion of filaments of the cyanobacteria Fluctiforma draycotensis. They find that filament motion consists of back and forth trajectories along a "track", interspersed with reversals of movement direction, with no clear dependence between filament speed and length. It is also observed that longer filaments can buckle and form plectonemes. A computational model is used to rationalize these findings.

Strengths:

Much work in this field focuses on molecular mechanisms of motility; by tracking filament dynamics this work helps to connect molecular mechanisms to environmentally and industrially relevant ecological behavior such as aggregate formation.

The observation that filaments move on tracks is interesting and potentially ecologically signifiant.

The observation of rotating membrane-bound protein complexes and tubular arrangement of slime around the filament provide important clues to the mechanism of motion.

The observation that long filaments buckle has potential to shed light on the nature of mechanical forces in the filaments, e.g. through study of the length dependence of buckling.

The comparison between motility on agar and on glass is interesting since it shows that filaments have both intrinsic propensity to reverse (that is seen on glass) and mechanically triggered reversal (that is seen on agar when the filament reaches the end of a track).

Weaknesses:

The manuscript makes the interesting statement that the distribution of speed vs filament length is uniform, which would constrain the possibilities for mechanical coupling between the filaments. However Fig 2C does not show a uniform distribution but rather an apparent lack of correlation between speed and filament length, although the statistical degree of correlation is not given. In my view, Fig 2C should not be described as a uniform distribution since mathematically that means something very different than what is shown here. Instead the figure should be described quantitatively with the use of a measured correlation coefficient. This also applies to Fig. S3A.

The statement "since filament speed results from a balance between propulsive forces and drag, these observations of no or positive correlation between filament speed and length show that all (or a fixed proportion of) cells in a filament contribute to propulsive force generation" helps to clarify the important link between Fig 2C and the concept that all cells contribute, but I think this statement is not obvious for many readers, and could be made clearer, e.g. by the use of a simple mathematical model for a chain of bacterial that accounts for drag forces and propulsion forces for each bacterium.

The authors have now clarified that the computational model is 1D and cannot explain the coupling between rotation, slime generation and motion. I find it encouraging and important that model predictions for the dwell time distributions (Fig S12 and S13) are similar to experimental measurements, but I think it would be better to put these results in the main text, together also with Fig S4. If these important results are in the supplement it is harder for the reader to assess the match between model and experiments.

Filament buckling is not analysed in quantitative detail, but the authors have now clarified that this will be the topic of future work with a 2D or 3D computational model.

---

## [Referee Report · Reviewer #2 (Public review)]

Summary:

The authors combined time-lapse microscopy with biophysical modeling to study the mechanisms and timescales of gliding and reversals in filamentous cyanobacterium Fluctiforma draycotensis. They observed the highly coordinated behavior of protein complexes moving in a helical fashion on cells' surfaces and along individual filaments as well as their de-coordination, which induces buckling in long filaments.

Strengths:

The authors provided concrete experimental evidence of cellular coordination and de-coordination of motility between cells along individual filaments. The evidence is comprised of individual trajectories of filaments that glide and reverse on surfaces as well as the helical trajectories of membrane-bound protein complexes that move on individual filaments and are implicated in generating propulsive forces.

Limitations:

The biophysical model is one-dimensional and thus does not capture the buckling observed in long filaments. I expect that the buckling contains useful information since it reflects the competition between bending rigidity, the speed at which cell synchronization occurs, and the strength of the propulsion forces.

Future directions:

The study highlights the need to identify molecular and mechanical signaling pathways of cellular coordination. In analogy to the many works on the mechanisms and functions of multi-ciliary coordination, elucidating coordination in cyanobacteria may reveal a variety of dynamic strategies in different filamentous cyanobacteria.

---

## [Referee Report · Reviewer #3 (Public review)]

Summary:

The authors present new observations related to the gliding motility of the multicellular filamentous cyanobacteria Fluctiforma draycotensis. The bacteria move forward by rotating their about their long axis, which causes points on the cell surface to move along helical paths. As filaments glide forward they form visible tracks. Filaments preferentially move within the tracks. The authors device a simple model in which each cell in a filament exerts a force that either pushes forwards or backwards. Mechanical interactions between cells cause neighboring cells to align the forces they exert. The model qualitatively reproduces the tendency of filaments to move in a concerted direction and reverse at the end of tracks.

The authors seek to understand how cells in a filament synchronize their motion to move in a concerted direction. This question connects to the evolution of multicellular life and so is important well beyond the specific field of cyanobacterial locomotion.

Strengths:

The biophysical model used to describe cell-cell coordination of locomotion is clear and reasonable. This model provides a useful phenomenological framework in which to consider the roles of individual cells in the coordinated motion of the group. The qualitative consistency between theory and observation suggests that this model captures some essential qualities of the true system.

The observation that filaments reverse at the ends of tracks is compelling, but difficult to clearly connect to any one microscopic model.

The observations of helical motion of the filament are compelling.

Weaknesses:

The comparison of theory and observation is mainly qualitative. While the authors have done a good job fitting the observations to the theory, it is not possible to systematically vary parameters, independently estimate parameter values, or apply external forces. Consequently, more experiments are needed before the proposed model can the accepted or rejected. This manuscript provides a promising hypothesis but not a compelling justification for it.

---

## [Author Response]

The following is the authors’ response to the original reviews

**Public Reviews:**

**Reviewer #1 (Public review):**
SummaryThe authors use microscopy experiments to track the gliding motion of filaments of the cyanobacteria Fluctiforma draycotensis. They find that filament motion consists of back-and-forth trajectories along a "track", interspersed with reversals of movement direction, with no clear dependence between filament speed and length. It is also observed that longer filaments can buckle and form plectonemes. A computational model is used to rationalise these findings.

We thank the reviewer for this accurate summary of the presented work.

Strengths:Much work in this field focuses on molecular mechanisms of motility; by tracking filament dynamics this work helps to connect molecular mechanisms to environmentally and industrially relevant ecological behavior such as aggregate formation.The observation that filaments move on tracks is interesting and potentially ecologically significant.The observation of rotating membrane-bound protein complexes and tubular arrangement of slime around the filament provides important clues to the mechanism of motion.The observation that long filaments buckle has the potential to shed light on the nature of mechanical forces in the filaments, e.g. through the study of the length dependence of buckling.

We thank the reviewer for listing these positive aspects of the presented work.

Weaknesses:The manuscript makes the interesting statement that the distribution of speed vs filament length is uniform, which would constrain the possibilities for mechanical coupling between the filaments. However, Figure 1C does not show a uniform distribution but rather an apparent lack of correlation between speed and filament length, while Figure S3 shows a dependence that is clearly increasing with filament length. Also, although it is claimed that the computational model reproduces the key features of the experiments, no data is shown for the dependence of speed on filament length in the computational model. The statement that is made about the model "all or most cells contribute to propulsive force generation, as seen from a uniform distribution of mean speed across different filament lengths", seems to be contradictory, since if each cell contributes to the force one might expect that speed would increase with filament length.

We agree that the data shows in general a lack of correlation, rather than strictly being uniform. In the revised manuscript, we intend to collect more data from observations on glass to better understand the relation between filament length and speed.

In considering longer filaments, one also needs to consider the increased drag created by each additional cell - in other words, overall friction will either increase or be constant as filament length increases. Therefore, if only one cell (or few cells) are generating motility forces, then adding more cells in longer filaments would decrease speed.

Since the current data does not show any decrease in speed with increasing filament length, we stand by the argument that the data supports that all (or most) cells in a filament are involved in force generation for motility. We would revise the manuscript to make this point - and our arguments about assuming multiple / most cells in a filament contributing to motility - clear.

The computational model misses perhaps the most interesting aspect of the experimental results which is the coupling between rotation, slime generation, and motion. While the dependence of synchronization and reversal efficiency on internal model parameters are explored (Figure 2D), these model parameters cannot be connected with biological reality. The model predictions seem somewhat simplistic: that less coupling leads to more erratic reversal and that the number of reversals matches the expected number (which appears to be simply consistent with a filament moving backwards and forwards on a track at constant speed).

We agree that the coupling between rotation, slime generation and motion is interesting and important when studying the specific mechanism leading to filament motion. However, we believe it is even more fundamental to consider the intercellular coordination that is needed to realise this motion. Individual filaments are a collection of independent cells. This raises the question of how they can coordinate their thrust generation in such a way that the whole filament can both move and reverse direction of motion as a single unit. With the presented model, we want to start addressing precisely this point.

The model allows us to qualitatively understand the relation between coupling strength and reversals (erratic vs. coordinated motion of the filament). It also provides a hint about the possibility of de-coordination, which we then look for and identify in longer filaments.

While the model’s results seem obvious in hindsight, the analysis of the model allows phrasing the question of cell-to-cell coordination, which so far has not been brought up when considering the inherently multi-cell process of filament motility.

Filament buckling is not analysed in quantitative detail, which seems to be a missed opportunity to connect with the computational model, eg by predicting the length dependence of buckling.

Please note that Figure S10 provides an analysis of filament length and number of buckling instances observed. This suggests that buckling happens only in filaments above a certain length.

We do agree that further analyses of buckling - both experimentally and through modelling would be interesting. This study, however, focussed on cell-to-cell coupling / coordination during filament motility. We have identified the possibility of de-coordination through the use of a simple 1D model of motion, and found evidence of such de-coordination in experiments. Notice that the buckling we report does not depend on the filament hitting an external object. It is a direct result of a filament activity which, in this context, serves as evidence of cellular de-coordination.

Now that we have observed buckling and plectoneme formation, these processes need to be analysed with additional experiments and modelling. The appropriate model for this process needs to be 3D, and should ideally include torques arising from filament rotation. Experimentally, we need to identify means of influencing filament length and motion and see if we can measure buckling frequency and position across different filament lengths. These works are ongoing and will have to be summarised in a separate, future publication.

**Reviewer #2 (Public review):**
Summary:The authors combined time-lapse microscopy with biophysical modeling to study the mechanisms and timescales of gliding and reversals in filamentous cyanobacterium Fluctiforma draycotensis. They observed the highly coordinated behavior of protein complexes moving in a helical fashion on cells' surfaces and along individual filaments as well as their de-coordination, which induces buckling in long filaments.

We thank the reviewer for this accurate summary of the presented work.

Strengths:The authors provided concrete experimental evidence of cellular coordination and de-coordination of motility between cells along individual filaments. The evidence is comprised of individual trajectories of filaments that glide and reverse on surfaces as well as the helical trajectories of membrane-bound protein complexes that move on individual filaments and are implicated in generating propulsive forces.

We thank the reviewer for listing these positive aspects of the presented work.

Limitations:The biophysical model is one-dimensional and thus does not capture the buckling observed in long filaments. I expect that the buckling contains useful information since it reflects the competition between bending rigidity, the speed at which cell synchronization occurs, and the strength of the propulsion forces.

Cell-to-cell coordination is a more fundamental phenomenon than the buckling and twisting of longer filaments, in that the latter is a consequence of limits of the former. In this sense, we are focussing here on something that we think is the necessary first step to understand filament gliding. The 3D motion of filaments (bending, plectoneme formation) is fascinating and can have important consequences for collective behaviour and macroscopic structure formation. As a consequence of cellular coupling, however, it is beyond the scope of the present paper.

Please also see our response above. We believe that the detailed analysis of buckling and plectoneme formation requires (and merits) dedicated experiments and modelling which go beyond the focus of the current study (on cellular coordination) and will constitute a separate analysis that stands on its own. We are currently working in that direction.

Future directions:The study highlights the need to identify molecular and mechanical signaling pathways of cellular coordination. In analogy to the many works on the mechanisms and functions of multi-ciliary coordination, elucidating coordination in cyanobacteria may reveal a variety of dynamic strategies in different filamentous cyanobacteria.

We thank the reviewer for highlighting this point again and seeing the value in combining molecular and dynamical approaches.

**Reviewer #3 (Public review):**
Summary:The authors present new observations related to the gliding motility of the multicellular filamentous cyanobacteria Fluctiforma draycotensis. The bacteria move forward by rotating their about their long axis, which causes points on the cell surface to move along helical paths. As filaments glide forward they form visible tracks. Filaments preferentially move within the tracks. The authors devise a simple model in which each cell in a filament exerts a force that either pushes forward or backwards. Mechanical interactions between cells cause neighboring cells to align the forces they exert. The model qualitatively reproduces the tendency of filaments to move in a concerted direction and reverse at the end of tracks.

We thank the reviewer for this accurate summary of the presented work.

Strengths:The observations of the helical motion of the filament are compelling. The biophysical model used to describe cell-cell coordination of locomotion is clear and reasonable. The qualitative consistency between theory and observation suggests that this model captures some essential qualities of the true system.The authors suggest that molecular studies should be directly coupled to the analysis and modeling of motion. I agree.

We thank the reviewer for listing these positive aspects of the presented work and highlighting the need for combining molecular and biophysical approaches.

Weaknesses:There is very little quantitative comparison between theory and experiment. It seems plausible that mechanisms other than mechano-sensing could lead to equations similar to those in the proposed model. As there is no comparison of model parameters to measurements or similar experiments, it is not certain that the mechanisms proposed here are an accurate description of reality. Rather the model appears to be a promising hypothesis.

We agree with the referee that the model we put forward is one of several possible. We note, however, that the assumption of mechanosensing by each cell - as done in this model - results in capturing both the alignment of cells within a filament (with some flexibility) and reversal dynamics. We have explored an even more minimal 1D model, where the cell’s direction of force generation is treated as an Ising-like spin and coupled between nearest neighbours (without assuming any specific physico-chemical basis). We found that this model was not fully able to capture both phenomena. In that model, we found that alignment required high levels of coupling (which is hard to justify except for mechanical coupling) and reversals were not readily explainable (and required additional assumptions). These points led us to the current, mechanically motivated model.

The parameterisation of the current model would require measuring cellular forces. To this end, a recent study has attempted to measure some of the physical parameters in a different filamentous cyanobacteria [1] and in our revision we will re-evaluate model parameters and dynamics in light of that study. We will also attempt to directly verify the presence of mechano-sensing by obstructing the movement of filaments.

**Summary from the Reviewing Editor:**
The authors present a simple one-dimensional biophysical model to describe the gliding motion and the observed statistics of trajectory reversals. However, the model does not capture some important experimental findings, such as the buckling occurring in long filaments, and the coupling between rotation, slime generation, and motion. More effort is recommended to integrate the information gathered on these different aspects to provide a more unified understanding of filament motility. In particular, the referees suggest performing a more quantitative analysis of the buckling in long filaments. Finally, it is also recommended to discuss the results in the context of previous literature, in order to better explain their relevance. Please find below the detailed individual recommendations of the three reviewers.

We thank the editor for this accurate summary of the presented work and for highlighting the key points raised by the reviewers. We have provided below point-by-point replies to these.

**Recommendations for the authors:**

**Reviewer #1 (Recommendations for the authors):**
The relevance of the study organism Fluctiforma draycotensis is not clearly explained, and the results are not discussed in the context of previous literature. The motivation would be clearer if the manuscript explained why this model organism was chosen and how the results compare with those previously observed for this or other organisms.

We have extended the introduction and discussion sections to make it clearer why we have worked with this organism and how the findings from this work relate to previous ones. In brief, *Flucitforma draycotensis* is a useful organism to work with as it not only displays significant motility but it also displays intriguing collective behaviour at different scales. Previous works on gliding motility in filamentous cyanobacteria have mostly focussed on the model organism *Nostoc punctiforme*, which only displays motility after differentiation into hormogonia [1]. There have also been studies in a range of different filamentous species, including those of the non-monophyletic genus, Phormidium, but these studies mostly looked at effects of genetic deletions on motility [2] or utilised electron microscopy to identify proteins (or surface features) involved in motility [3-5]. It must be noted that motility is also described and studied in non-filamentous cyanobacteria, but the dynamics of motion and molecular mechanisms there are different to filamentous cyanobacteria [6,7]. These previous studies are now cited / summarised in the revised introduction and discussion sections.

The inferred tracks, probably associated with secreted slime, play a key role since it is supposed that the tracks provide the external force that keeps the filaments straight. Movie S3, in phase contrast, provides convincing evidence for the tracks, but they cannot be seen in the fluorescence images presented in the main text. Clearer evidence of them should be shown in the main text. An especially important aspect of the tracks is where they start and end since the computational model assumes that reversal happens due to forces generated by reaching the end of a track. Therefore it seems important to comment on what produces the tracks, to check whether reversals actually happen at the end of a track, etc. Perhaps tracks could be strained with Concanavalin-A?

To confirm that reversals happen on track ends, we have now performed an analysis on agar, where we can see tracks on phase microscopy. This analysis confirms that, on agar, reversals indeed happen on track ends. We added this analysis, along with images showing tracks clearly as a new Fig in the main text (see new Fig. 1).

Further confirming the reversal at track ends, we note that filaments on circular tracks do not not reverse over durations longer than the ‘expected reversal interval’ of a filament on a straight track (see details in response to Reviewer 2).

Regarding what produces the tracks on agar, we are still analysing this using different methods and these results will be part of a future study. Fluorescent staining can be used to visualise slime tubes using TIRF microscopy, as shown in Fig. S8, however, visualising tracks on agar using low magnification microscopy has been difficult due to background fluorescence from agar.

We would also like to clarify that the model does not incorporate any assumptions regarding the track-filament interaction, other than that the track ends behave akin to a physical boundary for the filament. The observed reversal at track ends and “what” produces the track are distinct aspects of filament motion. We do not think that the model’s assumption of filament reversal at the end of the track requires understanding of the mechanism of slime production.

**Reviewer #3 (Recommendations for the authors):**
The manuscript combines three distinct topics: (1) the difference in locomotion on glass vs agar, (2) the development of a biophysical model, and (3) the helical motion of filament. It is not clear what insight one can gain from any one of these topics about the two others. The manuscript would be strengthened by more clearly connecting these three aspects of the work. A stronger comparison of theory to observation would be very useful. Some suggestions:(1) The observation that it is only the longest filaments that buckle is interesting. It should be possible to predict the critical length from the biophysical model. Doing so could allow fits of some model parameters.(2) What model parameters change between glass and agar? Can you explain these qualitative differences in motility by changing one model parameter?(3) Is it possible to exert a force on one end of a filament to see if it is really mechano-sensing that couples their motion?

We thank the reviewer for this comment and agree with them that a better connection between model and experiment should be sought. We believe that the new analyses, presented below in response to the 2nd suggestion of the reviewer, provide such a connection in the context of reversal frequency. As stated below, we think that the 1st suggestion falls outside of the scope of the current work, but should form the basis of a future study.

Regarding suggestion (1) - addressing buckling:

We agree with the reviewer that using a model to predict a critical buckling length would be useful. We note, however, that the presented study focussed on cell-to-cell coupling / coordination during filament motility using a 1D, beadchain model. The buckling observations served, in this context, as evidence of cellular de-coordination. Now that we have observed buckling (and plectoneme formation), these processes need to be analysed with further experiments and modelling. The appropriate model for studying buckling would have to be at least 2D (ideally 3D) and consider elastic forces and torques relating to filament bending, rotation, and twisting. Experimentally, we need to identify means of influencing filament length and motion and undertake further measurements of buckling frequency and position across different filament lengths. These investigations are ongoing and will be summarised in a separate, future publication.

Regarding suggestion (2) - addressing differences in motility on agar vs. glass:

We believe that the two key differences between agar and glass experiments are the occasional detachment of filaments from substrate on glass and the lack of confining tracks on glass. These differences might arise from the interactions between the filament, the slime, and the surface. As both slime and agar contain polysaccharides, the slime-agar interaction can be expected to be different from the slime-glass interaction. Additionally, in the agar experiments, the filaments are confined between the agar and a glass slide, while they are not confined on the glass, leaving them free to lift up from the glass surface. We expect these factors to alter reversal frequency between the two conditions. To explore this possibility, we have now extended the analysis of experimental data from glass and present that (see details below):

(i) dwell times are similar between agar and glass, and

(ii) reversal frequency distribution is different between glass and agar, and remains constant across filament length on glass.

We were able to explore these experimental findings with new model simulations, by removing the assumption of an “external bounding frame”. We then analysed reversal frequency within against model parameters, as detailed below.

“The movement of the filaments on glass. We have extended our analysis of motility on glass resulting in the following noted features. Firstly, the median speed shows a weak positive correlation with filament length on glass (see original Fig S3B vs. updated Fig. S3A). This is slightly different to agar, where we do not observe any strong correlation in either direction (see original, Fig. 1 vs. updated Fig 2). Both the cases of positive, and no correlation, support our original hypothesis that the propulsion force is generated by multiple cells within the filament.

Secondly, the filaments on glass display ‘stopping’ events that are *not* followed by a reversal, but are instead followed by a continuation in the original direction of motion, which we term ‘stop-go’ events, in contrast to the reversals. The dwell times associated with reversals and ‘stop-go’ events are similarly distributed (see original Fig S3A vs. updated Fig S3B). Furthermore, the dwell time distributions are similar between agar and glass (compare old Fig. 1C vs. new Fig 2C and new Fig. S3B). This suggests that the reversal process is the same on both agar and glass.

Thirdly, we find that the frequencies of both reversal and stop-go events on glass are uncorrelated with the filament length (see new Fig. S4A) and there are approximately twice as many reversals as stop-go events. In contrast, the filaments on agar reverse with a frequency that is inversely proportional to the filament length (which is in turn proportional to the track length) (see original Fig. S1). The distribution of reversal frequencies on agar is broader and flatter than the distribution on glass (see new Fig. S4B). These findings are inline with the idea that tracks on agar (which are defined by filament length) dictate reversal frequency, resulting in the strong correlations we observe between reversal frequency, track length, and filament length. On glass, filament movement is not constrained by tracks, and we have a specific reversal frequency independent of filament length.”

“Model can capture movement of filaments on glass and provides hypotheses regarding constancy of reversal frequency with length. We believe the model parameters controlling cellular memory (ω_max_) and strength of cellular coupling (K_ω_) describe the internal behaviour of a filament and therefore should not change depending on the substrate. Thus, we expect the model to be able to capture movement on glass just by removal of any ‘confining tracks’, i.e external forces, from the simulations. Indeed, we find that the model displays both stop-go and reversal events when simulated without any external force and can capture the dwell time distribution under this condition (compare new Figs. S12,S13 with S3).

In terms of reversal frequency, however, the model shows a reduction in reversal frequency with filament length (see new Fig. S15). This is in contrast to the experimental data. We find, however, that model results also show a reduction in reversal frequency with increasing ω_max_ and K_ω_ (see new Fig. S14 and S15). This effect is stronger with ω_max_, while it quickly saturates with K_ω_ (see new Fig. S14). Therefore, one possibility of reconciling the model and experiment results in terms of constant reversal frequency with filament length would be to assume that ω_max_ is decreasing with filament length (see new Fig. S16). Testing this hypothesis - or adding additional mechanisms into the model - will constitute the basis of future studies.”

Regarding suggestion (3) - role of mechanosensing:

We have tried several experiments to evaluate mechanosensing. First, we have used a micropipette or a thin wire placed on the agar, to create a physical barrier in the way of the filaments. The micropipette approach was not quite feasible in our current setup. The wire approach was possible to implement, but the wire caused a significant undulation / perturbation on agar. Possibly relating to this, filaments tended to continue moving alongside the wire barrier. Therefore, these experiments were inconclusive at this stage with regards to mechanosensing a physical barrier. As an alternative, we have attempted trapping gliding filaments using an optical trap with a far red laser that should not affect the physiology of the cells. This did not cause an immediate reversal in filament motion. However, this could be due to the optical trap strength being below the threshold value for mechanosensing. The force per unit length generated by filamentous cyanobacteria has been calculated via a model of self-buckling rods, giving a value of ≈1nN/μm [8]. In comparison, the optical trap generates forces on the scale of pN. Thus, the trap force is several orders of magnitude lower than the propulsive force generated by a filament, given filament lengths in the range of ten to several hundreds μm. We conclude that the lack of observed response may be due to the optical trap force being too weak.

Thus, the experiments we can perform using our current available methods and equipment are not able to prove either the presence or the absence of mechanosensing in the filament. We plan to perform further experiments in this direction, involving new and/or improved experimental setups, such as use of Atomic Force Microscopy.

We would like to note that there is an additional observation that supports the idea of reversals being mediated by mechanosensing at the end of a track, instead of the locations of the track ends being caused by the intrinsic reversal frequency of the filament. In a few instances (N = 4), filaments on agar ended up on a circular track (see Movie S4 for an example). These filaments did not reverse over durations a few times longer than the ‘expected reversal interval’ of a filament on a straight track.

Should $N$ following eq 7 and in eq 9 be $N_f$?

We have corrected this typo.

It would be useful to include references to what is known about mechanosensing in cyanobacteria.

We agree with the reviewer, and we have not updated the discussion section to include this information. Mechanosensing has not yet been shown directly in any cyanobacteria, but several species are shown to harbor genes that are implicated (by homology) to be involved in mechanosensing. In particular, analysis of cyanobacterial genomes predicts the presence of a significant number of homologues of the *Escherichia coli* mechanosensory ion channels MscS and MscL [9]. We have also identified similar MscS protein sequences in *F. draycotensis*. These channels open when the membrane tension increases, allowing the cell to protect itself from swelling and rupturing when subject to extreme osmotic shock. [10,11]

We also note that *F. draycotensis*, as with other filamentous cyanobacteria, have genes associated with the type IV pili, which may be involved in the surface-based motility [1]. Type IV pili have been shown to be mechanosensitive. For example, in cells of *Pseudomonas aeruginosa* that ‘twitch’ on a surface using type IV pili, application of mechanical shear stress results in increased production of an intracellular signalling molecule involved in promoting biofilm production. The pilus retraction motor has been shown to be involved in this shear-sensing response [12]. Additionally, twitching *P. aeruginosa* cells often reverse in response to collisions with other cells. Reversal is also caused by collisions with inert glass microfibres, which suggests that the pili-based motility can be affected by a mechanical stimulus [13].

References

(1) D. D. Risser, Hormogonium Development and Motility in Filamentous Cyanobacteria. Appl Environ Microbiol 89, e0039223 (2023).

(2) T. Lamparter et al., The involvement of type IV pili and the phytochrome CphA in gliding motility, lateral motility and photophobotaxis of the cyanobacterium Phormidium lacuna. PLoS One 17, e0249509 (2022)

(3) E. Hoiczyk, Gliding motility in cyanobacteria: observations and possible explanations. Arch Microbiol 174, 11-17 (2000).

(4) D. G. Adams, D. Ashworth, B. Nelmes, Fibrillar Array in the Cell Wall of a Gliding Filamentous Cyanobacterium. Journal of Bacteriology 181 (1999).

(5) L. N. Halfen, R. W. Castenholz, Gliding in a blue-green alga: a possible mechanism. Nature 225, 1163-1165 (1970).

(6) S. N. Menon, P. Varuni, F. Bunbury, D. Bhaya, G. I. Menon, Phototaxis in Cyanobacteria: From Mutants to Models of Collective Behavior. mBio 12, e0239821 (2021).

(7) F. D. Conradi, C. W. Mullineaux, A. Wilde, The Role of the Cyanobacterial Type IV Pilus Machinery in Finding and Maintaining a Favourable Environment. Life (Basel) 10 (2020).

(8) M. Kurjahn, A. Deka, A. Girot, L. Abbaspour, S. Klumpp, M. Lorenz, O. Bäumchen, S. Karpitschka Quantifying gliding forces of filamentous cyanobacteria by self-buckling. eLife 12:RP87450 (2024).

(9) S.C. Johnson, J. Veres, H. R. Malcolm, Exploring the diversity of mechanosensitive channels in bacterial genomes. Eur Biophys J 50, 25–36 (2021).

(10) S.I. Sukharev, W.J. Sigurdson, C. Kung, F. Sachs, Energetic and spatial parameters for gating of the bacterial large conductance mechanosensitive channel, MscL. Journal of General Physiology, 113(4), 525-540 (1999).

(11) N. Levina, S. Tötemeyer, N.R. Stoke, P. Louis, M.A. Jones, I.R. Boot. Protection of *Escherichia coli* cells against extreme turgor by activation of MscS and MscL mechanosensitive channels: identification of genes required for MscS activity. The EMBO journal (1999).

(12) V.D. Gordon, L. Wang, Bacterial mechanosensing: the force will be with you, always. Journal of cell science 132(7):jcs227694 (2019).

(13) M.J. Kühn, L. Talà, Y.F. Inclan, R. Patino, X. Pierrat, I. Vos, Z. Al-Mayyah, H. Macmillan, J. Negrete Jr, J.N. Engel, A. Persat, Mechanotaxis directs *Pseudomonas aeruginosa* twitching motility. Proceedings of the National Academy of Sciences. 118(30):e2101759118 (2021).